# Mucosal Mast Cells as Key Effector Cells in Food Allergies

**DOI:** 10.3390/cells11030329

**Published:** 2022-01-19

**Authors:** Nobuhiro Nakano, Jiro Kitaura

**Affiliations:** Atopy (Allergy) Research Center, Juntendo University Graduate School of Medicine, Tokyo 113-8421, Japan; j-kitaura@juntendo.ac.jp

**Keywords:** mucosal mast cells, notch signaling, TGF-β, food allergy, oral immunotherapy

## Abstract

Mucosal mast cells (MMCs) localized in the intestinal mucosa play a key role in the development of IgE-mediated food allergies. Recent advances have revealed that MMCs are a distinctly different population from connective tissue mast cells localized in skin and other connective tissues. MMCs are inducible and transient cells that arise from bone marrow-derived mast cell progenitors, and their numbers increase rapidly during mucosal allergic inflammation. However, the mechanism of the dramatic expansion of MMCs and their cell functions are not well understood. Here, we review recent findings on the mechanisms of MMC differentiation and expansion, and we discuss the potential for the inducers of differentiation and expansion to serve as targets for food allergy therapy. In addition, we also discuss the mechanism by which oral immunotherapy, a promising treatment for food allergy patients, induces unresponsiveness to food allergens and the roles of MMCs in this process. Research focusing on MMCs should provide useful information for understanding the underlying mechanisms of food allergies in order to further advance the treatment of food allergies.

## 1. Introduction

Humans have long suffered from food allergies. Hippocrates (460-377 BC), an ancient Greek physician, wrote about a suspected case of allergy to cheese [1]. The principal treatment for food allergies is the identification and avoidance of the causative food, which has remained largely unchanged from ancient times to the present. However, these approaches create a serious problem with food energy intake [2]. Therefore, the development of therapeutic agents and treatments for food allergies is in high demand, but to achieve this, we need to understand the underlying mechanisms.

Food allergies are classified into three groups according to the immunological mechanism: IgE-mediated, non-IgE-mediated, and mixed. Typical food allergies are IgE-mediated, and thus the main symptoms are caused by immediate hypersensitivity reactions in which food allergen-specific IgE and mast cells play a central role. In some non-IgE-mediated allergies, such as pseudo-allergies, mast cell activation also plays an important role in the onset of allergic symptoms [3,4,5]. In developed countries, the prevalence of food allergies is 6% to 10% in infants and 2% to 5% in adults [6,7,8,9], indicating that they are more common in children. However, it has recently been demonstrated that some patients with irritable bowel syndrome (IBS) have abdominal pain caused by local IgE-mediated food allergy reactions [10]. The prevalence of IBS in developed countries is approximately 10% to 15% and is more common in adults. More than 50% of patients with IBS have symptoms triggered by a meal, despite negative results of serum food allergen-specific IgE antibodies and skin prick tests [11,12]. Recent studies have shown that many of these patients are negative for food allergen-specific IgE in serum but positive for the intestinal mucosa, which leads to a local IgE-mediated food allergy reaction [10,12]. Based on these findings, Aguilera-Lizarraga et al. [10] proposed a new disease concept called “food-induced disorders mediated by mast cell activation”. IBS, in which foods are involved in the onset of symptoms, is part of a spectrum of food-induced disorders, and systemic food allergy is at extreme end of the spectrum.

Many studies using animal models have shown that mast cells in the intestinal mucosa are the major effector cells in IgE-mediated food-induced disorders, including food allergies [10,13,14,15,16]. Therefore, intestinal mucosal mast cells are an attractive target for therapeutic intervention in those diseases. In this review, we summarize the differentiation mechanisms and functions of intestinal mucosal mast cells and discuss the possibility of treating food-induced disorders by regulating them.

## 2. Mucosal Mast Cells

Mast cells are categorized into two types according to the staining patterns and anatomical locations: connective tissue-type and mucosal-type (Table 1).

### 2.1. Mouse Mast Cell Subtypes

Mouse connective tissue mast cells (CTMCs), which are located around venules and nerve endings in most connective tissues, contain heparin in the granules and are therefore stained metachromatically by dyes such as toluidine blue, alcian blue, safranin O, and berberine sulphate [18,23,24,29,30,31,32]. In addition, CTMCs are characterized by the expression of mouse mast cell proteases (mMCP)-4, -5, -6, and -7, as well as carboxypeptidase A3 (CPA3). In contrast, mouse mucosal mast cells (MMCs), which are located inside the epithelia of the intestinal and respiratory mucosa, lack heparin in the granules and are difficult to detect with standard histological staining protocols [18,29,30,31,32]. However, the expression of mMCP-1 and -2 is a characteristic feature of MMCs, and these may be appropriate markers for immunohistochemical staining. In addition, MMCs express αE-integrin/CD103 on the cell surface [18,19,20]. The αE integrin is paired with the β7 integrin to form the heterodimer αEβ7 that binds to epithelial-associated E-cadherin [33]. The expression of this adhesion molecule may contribute to the accumulation of MMCs in the mucosal tissues.

Furthermore, the lifespan is notably different between CTMCs and MMCs. This may be based on differences in the progenitors of each subtype. CTMCs are long-lived tissue-resident cells derived from fetal-derived progenitors and maintained locally by self-proliferation [19,21]. However, MMCs are inducible and transient cells that arise from bone marrow-derived mast cell progenitors (MCp), which have been identified as lineage^−^c-Kit^+^Sca-1^−^Ly6c^−^FcεRIα^−^CD27^−^β7-integrin^+^ST2^+^ [32,34]. Taken together, MMCs are a distinctly different population from CTMCs; however, their differentiation process and detailed functions are not fully understood.

### 2.2. Human Mast Cell Subtypes

In humans, mast cells expressing both tryptase and chymase (MC_TC_) and mast cells expressing tryptase without chymase (MC_T_) are considered to correspond to mouse CTMCs and MMCs, respectively [19]. Although the distribution of human mast cell subtypes is not as strictly restricted as in mice, MC_T_s constitute approximately 80% of mast cells in the intestinal mucosa [35]. In addition, proteome analyses have shown that human and mouse mast cell functions are evolutionarily conserved and very similar [36]. Intestinal mast cells are found throughout the gastrointestinal tract and normally account for 2–3% of mononuclear cells in the lamina propria. Ehrsam et al. [37] have recently reported the number and distribution of mast cells in the pediatric gastrointestinal tract in healthy individuals, as well as in patients with food allergies. Importantly, mast cell hyperplasia and higher tryptase concentrations in the intestinal mucosa are associated with allergic inflammation of the mucosa in patients with food allergies and IBS [38,39]. However, since a committed human MCp, which is identified as CD34^+^β7-integrin^+^, is found in both adult peripheral blood [27] and fetal cord blood [28], it is currently unclear how MC_TC_s and MC_T_s are maintained in each tissue.

Taken together, the activation of intestinal mucosal mast cells (mouse MMCs and human MC_T_) plays a key role in the development of IgE-mediated food allergies.

### 2.3. Mediators Released by Activated MMCs

In the immediate phase of IgE-mediated food allergy, MMCs function as major effector cells through the IgE-dependent release of a wide variety of chemical mediators, such as biogenic amine, lipid mediators, proteases, and inflammatory cytokines [40,41]. Histamine, which is a major amine released from activated mast cells, leads to vasodilation for the recruitment of leukocytes, which in some cases triggers anaphylaxis [41]. In addition, histamine causes abdominal pain by binding histamine receptors on sensory neurons in the gut [10]. Serotonin, also a biogenic amine released from activated mast cells, contributes to the pathophysiology of anaphylaxis and allergic diarrhea [13,41]. The lipid mediator platelet-activating factor (PAF) also contributes to the induction of anaphylaxis and allergic diarrhea [13,41]. Prostaglandin D_2_ (PGD2), a lipid mediator produced in large quantities by mast cells, accelerates type 2 inflammation by accumulating inflammatory cells in mouse models of asthma [42]. In contrast, in a mouse model of IgE-mediated food allergies, the deficiency of PGD2 results in mast cell hyperplasia and exacerbates food allergen-induced allergic reactions [16]. Activated MMCs immediately release MMC-specific proteases mMCP-1 and mMCP-2 (in humans, tryptase is released from MC_T_). Although the effects of these proteases on the pathogenesis of food allergies are unknown, they are much more stable in the blood than histamine and are useful as markers of MMC or MC_T_ activation [13,43].

The late-phase allergic response, which occurs several hours after the immediate phase, is characterized by the infiltration of inflammatory cells into tissues. The migration of these cells is stimulated by cytokines released from activated MMCs in the immediate phase. Eosinophils are mobilized by IL-5 released by mast cells, and neutrophils and lymphocytes are attracted by TNF-α, causing inflammation and tissue destruction [40,44]. IL-13 is involved in Th2 development, B cell proliferation and isotype switching, and goblet cell hyperplasia with mucus hypersecretion. Importantly, it has been demonstrated that IL-13 released by mast cells contributes to the pathogenesis of IgE-mediated food allergies in mice [45].

Mas-related G protein-coupled receptor-X2 (MRGPRX2) (murine ortholog, Mrgprb2) is known as a receptor that causes IgE-independent mast cell activation. The receptor recognizes a broad range of cationic ligands, including certain types of drugs, host defense peptides (such as LL-37), and neuropeptides [3,5]. Importantly, the receptor is expressed on CTMCs, but not on MMCs (Table 1). Thus, pseudo-allergic reactions triggered by these molecules, even those ingested orally, may differ from IgE-dependent food allergies in the profile of secreted mediators.

### 2.4. Biomarkers of Food Allergies Derived from Activated Mast Cells

As mentioned above, multiple mediators are released from activated mast cells, including MMCs, during the effector phase of IgE-mediated food allergies. Some of these mediators are useful as biomarkers for objectively detecting allergic reactions in patients with food allergies. The biomarkers in blood or urine are important for the accurate detection of allergic reactions caused by oral food challenge, which is the gold standard for the diagnosis of food allergies and for the clinical diagnosis of anaphylaxis. Histamine is a typical mediator released by IgE-mediated mast cell degranulation. However, in order to properly measure plasma histamine levels, it is necessary to obtain a blood sample within 15 and 60 min after the onset of anaphylactic symptoms. In addition, special handling of the blood sample is required [46]. In contrast, urinary histamine and its metabolite, N-methylhistamine, can be measured in samples obtained 24 h after the onset of symptoms [46,47]. Tryptase is also a useful mediator for diagnosing of anaphylaxis. In particular, serum total tryptase levels are strongly associated with insect sting-triggered anaphylaxis [48,49]. However, tryptase levels are lower in MMCs (or MC_T_) than in cutaneous and perivascular CTMCs (or MC_TC_); thus, anaphylactic reactions and increased tryptase levels elicited by food allergies may be weaker than those elicited by insect sting or intravenous drugs [50]. Recently, a urinary PGD2 metabolite, 11,15-dioxo-9α-hydroxy-2,3,4,5-tetranorprostan-1,20-dioic acid, was reported to be a useful diagnostic index of food allergies in both mice and humans [51,52]. Moreover, recent metabolomic analyses reveal many promising candidates for food allergy biomarkers [53]. Crestani et al. [54] show that multifood allergies are associated with tryptophan, an important metabolite in the production of serotonin, and 1-arachidonoyl-GPD, a metabolite upstream of PAF. These biomarkers are released at different times after the activation of mast cells. Therefore, it is desirable to have many biomarkers that can be used for diagnosis.

## 3. Inducers of MMC Differentiation

### 3.1. Cytokines

In an experimental IgE-mediated food allergy mouse model, intestinal MMCs are dramatically and rapidly expanded by repeated oral administration of food allergens to mice sensitized to the allergen. MMCs arise from bone marrow-derived MCp that are recruited to mucosal tissues in response to T cell-mediated type 2 allergic inflammation, such as IgE-mediated food allergy [32,34]. The differentiation of MCp into mature MMCs in situ indicates that there are some inducers of MMC differentiation in the mucosal tissues. Transforming growth factor-β (TGF-β) is known to induce the expression of MMC-specific proteases, mMCP-1 and -2, and the cell surface CD103 in in vitro culture conditions [20,55,56,57]. Single-cell RNA sequencing of MMCs isolated from mouse models of allergic lung inflammation strongly suggests the involvement of TGF-β in the differentiation of MMCs in vivo [32]. Therefore, TGF-β is a promising candidate as an MMC differentiation inducer (Figure 1). In addition, it has also been reported that IL-10 induces MMC-specific protease expression in bone marrow-derived mast cells (BMMCs) [58,59] (Figure 1). TGF-β and IL-10 are important cytokines for the differentiation and maintenance of various immune cells belonging to the mucosal immune system, such as CD103^+^ dendritic cells and regulatory T cells (Treg) [60,61,62]. This implies that MMCs are also members of the mucosal immune system.

### 3.2. Notch Ligands

In addition to those cytokines, we previously showed that Notch ligands are potent inducers of MMC differentiation [63] (Figure 1). In mammals, four Notch receptors, Notch1–4, which are epidermal growth factor-like transmembrane receptors, have been identified and are activated upon interaction with four transmembrane ligands (Jagged1, Jagged2, Delta-like 1 (DLL1), and DLL4). Notch receptor-mediated signaling (Notch signaling) is transmitted into the nucleus of recipient cells through cell–cell contact and regulates various cell fate determinations, such as lymphocyte development [64,65]. Mouse mast cells constitutively express Notch1 and Notch2 on the cell surface [66]. We found that when mouse BMMCs, immature mast cells, were co-cultured with a CHO cell line expressing the Notch ligand, the expression of MMC markers was induced in BMMCs. Notably, the gene expression profile of intestinal MMCs isolated from the small intestine of naive mice was closer to MMC-like cells generated in the presence of a Notch ligand than MMC-like cells generated in the presence of TGF-β. Moreover, the co-culture of BMMCs with cells comprising the mouse intestinal mucosa, which express Notch ligands in some cells, induced the expression of MMC markers in BMMCs in a Notch signaling-dependent manner [63]. Therefore, signals provided by TGF-β, IL-10, and Notch ligands are predicted to be involved in MMC differentiation in the intestinal mucosa.

## 4. Molecular Mechanisms of MMC Differentiation

Canonical TGF-β signaling is mediated by SMAD family proteins, which translocate to the nucleus and act as transcription factors. *Mcpt1* and *Mcpt2* genes, which encode MMC-specific proteases mMCP-1 and mMCP-2, respectively, have SMAD-responsive motifs and GATA motifs upstream of them. The GATA motif is a consensus nucleotide sequence to which GATA transcription factors bind directly. Kasakura et al. [57] have revealed that TGF-β induces the transcription of *Mcpt1* and *Mcpt2* genes by mobilizing SMAD4 and GATA2 to those elements and motifs in BMMCs. In addition, the human *ITGAE* gene, which encodes CD103, has SMAD-responsive motifs and NFAT binding sites on both the promoter and enhancer. TGF-β induces the transcription of the *ITGAE* gene by mobilizing SMAD3 and the transcription factor NFAT-1 to the motifs and sites in human CD8^+^ T cells [67]. Thus, TGF-β directly induces the transcription of the MMC marker genes, suggesting that it plays a central role in the differentiation of MMCs.

So how does Notch signaling induce MMC differentiation? What is the relationship between Notch and TGF-β signaling? To address these questions, we studied the molecular mechanisms by which Notch signaling induces MMC differentiation using mouse BMMCs. We first showed that the expression levels of MMC markers were synergistically enhanced in BMMCs by a combination of Notch ligand and TGF-β. Interestingly, Notch-mediated MMC marker expression was inhibited by a selective inhibitor of the TGF-β type I receptor (i.e., a TGF-β signaling inhibitor), and TGF-β-mediated MMC marker expression was inhibited by a γ-secretase inhibitor (i.e., a Notch signaling inhibitor). These results indicate that Notch and TGF-β signaling function interdependently in the expression of MMC markers. In addition, Notch-mediated MMC marker expression in BMMCs was suppressed by the knockdown of SMAD4 expression using siRNA. Therefore, Notch signaling induces marker expression in a SMAD-dependent manner. However, Notch signaling did not upregulate TGF-β production, TGF-β receptor expression, or phosphorylation of SMADs in BMMCs, indicating that Notch signaling does not directly enhance TGF-β signaling. We found that SMADs were always weakly phosphorylated in BMMCs by TGF-β contained in the medium, even without the addition of TGF-β. This was confirmed by the fact that the removal of TGF-β from the medium significantly suppressed the Notch-mediated transcription of MMC marker genes. These results suggest that pericellular TGF-β is required for Notch-mediated transcription of MMC marker genes and that Notch signaling acts to promote TGF-β/SMAD signaling-dependent gene transcription. Next, by chromatin immunoprecipitation assays, we showed that Notch signaling significantly increased the levels of active histone marks at the promoter regions of *Mcpt1* and *Mcpt2*. Furthermore, the nuclear accumulation of SMAD3 and SMAD4 was observed in BMMCs that were forced to express the active Notch2 intracellular domain, which translocated to the nucleus to form a transcription complex. Taken together, Notch and TGF-β signaling play synergistic and interdependent roles in inducing the differentiation of MMCs (Figure 2) [68]. These roles may contribute to the rapid expansion of the number of MMCs during allergic mucosal inflammation.

## 5. Inducers of MMC Expansion

MMC hyperplasia in the intestinal mucosa in IgE-mediated food allergies probably involves both the recruitment of MCps and in situ MMC proliferation. Thus, MMC growth factors are thought to be abundant in inflamed mucosal tissues. In particular, IL-3, IL-4, IL-9, and stem cell factor (SCF) are known as major growth factors of MMCs (Figure 1). The survival and proliferation of mouse MMCs depends on the presence of IL-3. Since IL-3 is produced primarily by T cells, MMC proliferation is, therefore, T-cell-dependent [69]. In addition, IL-3-dependent MMC proliferation is further enhanced by IL-4 [70]. SCF, a ligand for c-Kit produced mainly by stromal cells and epithelial cells, is also an important growth factor for both mouse and human mast cells. It has been reported that SCF is critical for the development of MMC hyperplasia in murine models of food allergy and infection with intestinal parasites [71,72]. In addition, IL-4 markedly amplifies the SCF-dependent proliferation of human mast cells isolated from intestinal tissues [73]. Moreover, IL-4 has been shown to not only enhance IL-3- or SCF-dependent mast cell proliferation, but also to directly stimulate proliferation via STAT6 in a mouse mode of food allergies [74]. In food allergies, IL-4 is produced by basophils and IL-33-stimulated group 2 innate lymphoid cells (ILC2s) in addition to Th2 cells [75,76]. Abundant IL-4 stimulates IgE production by B cells and inhibits Treg functions, resulting in enhanced MMC activation. In humans, data from single-cell RNA sequencing analysis show a significant correlation between the number of mast cells in the tissue and the expression levels of IL-4-related genes. Moreover, IL-4 has also been shown to confer an inflammatory phenotype on MC_T_ [77]. Therefore, IL-4 is a key determinant of MMC expansion in food allergies.

In addition to those cytokines, IL-9, previously described as mast cell growth-enhancing activity (MEA), is also an important cytokine that induces MMC hyperplasia in a mouse model of food allergy [78]. Although a well-known source of IL-9 is helper T cells, it has recently been shown that a population of MMCs, termed IL-9-producing MMCs (MMC9s), is a critical source of IL-9 in IgE-mediated food allergy [79,80]. MMC9s are generated from bone marrow-derived MCp in the presence of IL-4 and produce prodigious amounts of IL-9 in response to IL-33 released from injured epithelial cells, leading to MMC expansion.

Taken together, IL-3 and SCF may play a fundamental role in mast cell survival and proliferation. IL-4 enhances the effects of those cytokines while conferring an inflammatory phenotype on MMCs. Inflammatory MMCs actively produce IL-9 to induce MMC expansion, resulting in increased susceptibility to IgE-mediated food allergy.

## 6. MMCs as a Potential Therapeutic Target for the Treatment of Food Allergies

### 6.1. Blockade of Notch Signaling

Notch signaling is a potent inducer of MMC differentiation. To investigate whether the symptoms of food allergies are attenuated by the inhibition of Notch signaling, we administered the Notch signaling inhibitor DAPT, also known as a γ-secretase inhibitor, to a mouse model of IgE-meditated food allergy sensitized with ovalbumin (OVA). The expansion of MMCs in the small intestine and colon was significantly suppressed in mice treated with DAPT during the effector phase. Predictably, the severity of allergic diarrhea and the degree of hypothermia induced by oral administration of OVA were lower in mice treated with DAPT than in the control mice. The concentration of serum OVA-specific IgE in DAPT-treated mice was the same level as in the control mice, indicating that the reduction in MMC numbers contributed to the attenuation of symptoms [63]. Since Notch signaling is not only involved in MMC differentiation but also in the differentiation of various immune cells, long-term administration of a Notch signaling inhibitor may lead to undesirable side effects in food-allergic patients. Therefore, the idea of administering a Notch signaling inhibitor to patients with food allergies is not very promising. However, these results suggest that the inhibition of MMC differentiation and the suppression of Notch target gene expression in mast cells can be effective in the treatment of IgE-mediated food allergies.

### 6.2. Blockade of IL-4 Signaling

In allergic inflammation in both human and mouse mucosa, IL-4 is a key cytokine for inducing MMC expansion and conferring inflammatory phenotypes to MMCs. Thus, the blockade of IL-4 signaling may prevent MMC expansion in IgE-mediated food allergies. Dupilumab, a fully human monoclonal antibody against the IL-4 receptor α chain (IL-4Rα), has already been used to treat patients with severe atopic dermatitis (AD), asthma, and chronic rhinosinusitis with nasal polyposis (CRSwNP) [81]. IL-4Rα is shared with IL-4 receptor and IL-13 receptor, and thus dupilumab blocks both IL-4 and IL-13 signaling. IL-4 and IL-13 are signature type 2 cytokines and exert diverse effects on multiple cells. IL-4 plays a critical role in Th2 differentiation in T cells and IgE class switching in B cells. IL-13 induces goblet cell hyperplasia and smooth muscle cell proliferation [82]. In addition, both IL-4 and IL-13 act on mast cells, basophils, eosinophils, macrophages, fibroblasts, epithelial cells, and keratinocytes [83]. Dwyer et al. [77] performed data analysis of single-cell RNA sequencing from a patient with CRSwNP evaluated before and after six weeks of treatment with dupilumab and found markedly reduced expression of *IL17RB* (encoding IL-25 receptor), a newly identified marker of human MC_T_s. This indicates that dupilumab also targets MMCs and may be effective in suppressing mucosal allergy inflammation caused by MMCs. At this time, there are only a few reports of food-allergic patients treated with dupilumab. Rial et al. [84] have recently reported that a 30-year-old woman with severe AD who had newly developed a food allergy to corn and nuts acquired a tolerance to these foods by treatment with dupilumab. This case report suggests that the blockade of IL-4 signaling may be effective in the treatment of food allergies, although it is unclear whether MMCs are the primary target.

IL-33 and IL-25 secreted by various cell types activate Th2-type cytokine production and exacerbate allergic diseases. The neutralization of these cytokines by specific antibodies has been shown to be effective in suppressing the Th2 immune responses [85]. Therefore, these cytokines, which enhance the production of IL-4 and IL-13, may also be potent therapeutic targets.

### 6.3. Activation of an Inhibitory Receptor

Mast cells express the inhibitory receptor CD300f, also known as leukocyte mono-immunoglobulin-like receptor 3 (LMIR3), on the cell surface. CD300f has two immunoreceptor tyrosine-based inhibitory motifs (ITIMs) and an immunoreceptor tyrosine-based switch motif (ITSM) in its intracellular region, which are phosphorylated upon binding to its ligand ceramide [86]. Src homology 2 domain-containing protein phosphatase (SHP)-1 and -2, which are recruited to the phosphorylated ITIMs and ITSM, negatively regulate high-affinity IgE receptor FcεRI-mediated signaling, resulting in the suppression of IgE-mediated mast cell activation. Uchida et al. [87] have shown that CD300f is also expressed on the cell surface of MMCs, MMC9, and β7 integrin^+^ MCp in the small intestinal lamina propria in mice. In a mouse model of IgE-mediated food allergy, the serum level of mMCP-1, a marker of MMC activation, was higher in CD300f-deficient mice than in wild-type mice after oral allergen challenge. Moreover, food allergy responses, such as allergic diarrhea and the transient drop in body temperature induced by allergen challenge, were exacerbated in CD300f-deficient mice compared to wild-type mice. These results suggest that IgE/MMC-mediated food allergy reactions may be attenuated by the activation of CD300f. Based on this hypothesis, ceramide-containing vesicles were administrated to the mouse model of IgE-mediated food allergy. As expected, the food allergy responses were attenuated in mice pretreated with the vesicles. Therefore, CD300f is a potential therapeutic target for the treatment of IgE-mediated food allergies.

In addition to the inhibitory receptor, inhibitory cytokines, such as IL-37, may also function to suppress excessive activation of mast cells [88]. Thus, these natural inhibitors of immune responses could also be used as therapeutic agents.

## 7. Roles of MMCs in Oral Immunotherapy

Oral immunotherapy (OIT) is a promising treatment for patients with food allergies to induce desensitization and eventually sustain unresponsiveness to food allergens. In general, the process of OIT is composed of an initial build-up in which the allergen is administered repeatedly with a gradual increase followed by a maintenance phase. The first goal of OIT is desensitization, which is defined as a lack of clinical reactivity to allergens and is maintained by regular allergen exposure. The efficacy of OIT for egg, milk, and peanuts has been demonstrated by many randomized placebo-controlled trials [89]. However, OIT also has the risks of adverse reactions, such as oral itching, eosinophilic gastrointestinal disorders, and anaphylaxis [89,90]. To mitigate the risks of OIT, there is great interest in developing approaches to effectively and safely induce desensitization.

### 7.1. Harmful Effects of MMCs

Undesirable allergic reactions during OIT are triggered by the IgE-mast cell axis. One approach to prevent these reactions is to use an anti-IgE antibody. Omalizumab is a humanized IgG1κ monoclonal antibody that binds to the Fc region of human IgE and is currently applied for the treatment of severe uncontrolled asthma, allergic rhinitis, and chronic spontaneous urticaria [91,92,93]. This antibody blocks the interaction between IgE and FcεRI and reduces the level of free IgE in circulation, preventing the development of IgE-mediated allergic reactions. The use of omalizumab with OIT has been investigated in several small trials for the treatment of milk, egg, and peanut allergies [94]. Notably, these trials have shown that the combination of omalizumab and OIT can not only reduce adverse events but also the time to acquisition of desensitization. These reports support the idea that the activation of mast cells, including MMCs, impairs the acquisition of desensitization to allergens. Therefore, blockade of the IgE-mast cell axis will improve the safety and efficacy of OIT.

### 7.2. Helpful Effects of MMCs

There are several reports on the mechanisms by which the early initiation phase of OIT induces desensitization. Displacement of the actin cytoskeleton in desensitized mast cells inhibits IgE-dependent calcium influx and thus suppresses mast cell degranulation [95]. Another report demonstrates that OIT-induced mast cell hyporesponsiveness is induced by enhanced internalization of IgE bound to the cell surface [96]. These studies indicate that mast cells become unresponsive to allergens, resulting in the suppression of the induction of allergic symptoms. In addition, it is believed that the expansion of the Treg population is involved in the acquisition of desensitization and/or sustained unresponsiveness to allergens in the late initiation phase of OIT [97,98]. The mechanism by which Treg expansion is induced in cases of successful OIT is poorly understood. Previous studies have shown that, under certain conditions, mast cells play a role in immunosuppression by producing cytokines, such as IL-2 and IL-10 [99,100]. IL-2 is essential for the proliferation, survival, maintenance, and functional capacity of Tregs, which constitutively express IL-2 receptor α chain (CD25) [101]. Although IL-10 is a well-known immunosuppressive cytokine produced by Tregs, it also plays an important role in the differentiation and functional capacity of Tregs themselves [102,103]. Takasato et al. [104,105] have demonstrated a helpful effect of MMCs in OIT using a mouse model of OIT. OIT shifted MMCs from pathogenic (or pro-allergic) status to regulatory (desensitization) status. OIT-induced desensitized MMCs contributed to the expansion of Tregs by releasing immunosuppressive cytokines, such as IL-2 and IL-10, in addition to reduced reactivity to allergens and the suppressed release of the Th2 cytokine, IL-4. These results provide new insights into the role of MMCs in the mucosal immune system.

The second goal of OIT is the acquisition of sustained unresponsiveness to allergens, which is the long-term loss of reactivity to allergens independent of continued exposure to them. However, current OIT therapy has not been able to efficiently induce sustained unresponsiveness for specific allergens to food-allergic patients [89,106]. A major problem is that the mechanism by which sustained unresponsiveness is induced by OIT remains unknown. We have recently shown that the expansion of immunosuppressive cells, including Tregs, IL-10-expressing Th2, and myeloid-derived suppressor cells (MDSCs), contributes to the establishment of sustained unresponsiveness by OIT, and that Notch signaling is involved in these expansion processes [107]. However, the roles of MMCs were not analyzed in this study, and thus further research is needed.

## 8. Concluding Remarks

In summary, intestinal MMCs play a central role in the development of food-induced disorders mediated by mast cell activation, including systemic food allergies. Mouse MMCs are induced to differentiate mainly by TGF-β and Notch ligands, stimulated to proliferate rapidly by IL-4 and IL-9, and converted to an inflammatory phenotype by IL-4. Although there are still many unanswered questions about human mast cells, transcriptome and proteome analyses suggest that they are not so different from mouse mast cells [36]. Thus, these cytokines and ligands are potential therapeutic targets for food-induced disorders.

Recent advances have revealed that MMCs (MC_T_ in humans) are a distinctly different population from CTMCs (MC_TC_ in humans). However, the functions of MMCs are still poorly understood. The number of MMCs increases dramatically during mucosal allergic inflammation. However, are β7 integrin^+^ MCp the only source of MMCs? What happens to the expanded MMCs after the inflammation has ended? Food allergies in humans are often accompanied by cutaneous symptoms in addition to mucosal symptoms. Are MMCs involved in the induction of cutaneous symptoms such as erythema and urticaria? Food-induced disorders, including systemic food allergy, cause abdominal pain. How do MMCs and sensory nerves interact? Do MMCs contribute to the establishment of oral tolerance? MMCs may have as-yet-unidentified functions and roles that are unique to the mucosal immune system. Therefore, research focusing on MMCs will further advance the treatment of food-induced disorders.

## Figures and Tables

**Figure 1 cells-11-00329-f001:**
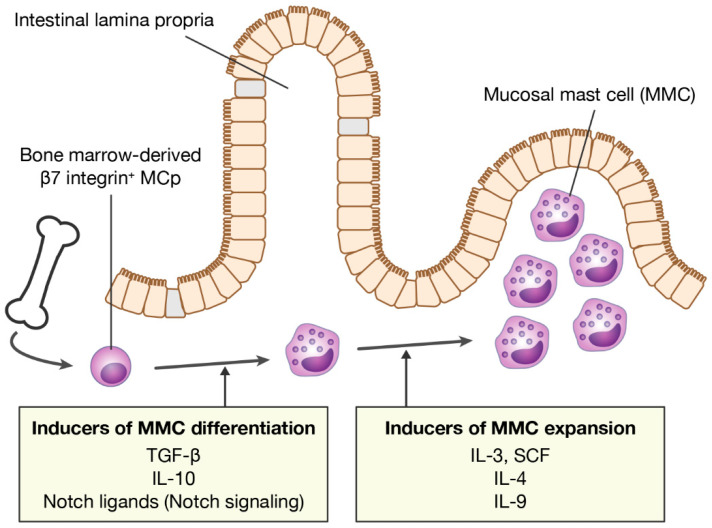
Inducers of MMC differentiation and expansion. MMCs differentiate from bone marrow-derived MCps in mucosal tissues. MMC numbers increase rapidly during allergic inflammation. MMC, mucosal mast cell; MCp, mast cell progenitor; TGF-β, transforming growth factor-β; SCF, stem cell factor.

**Figure 2 cells-11-00329-f002:**
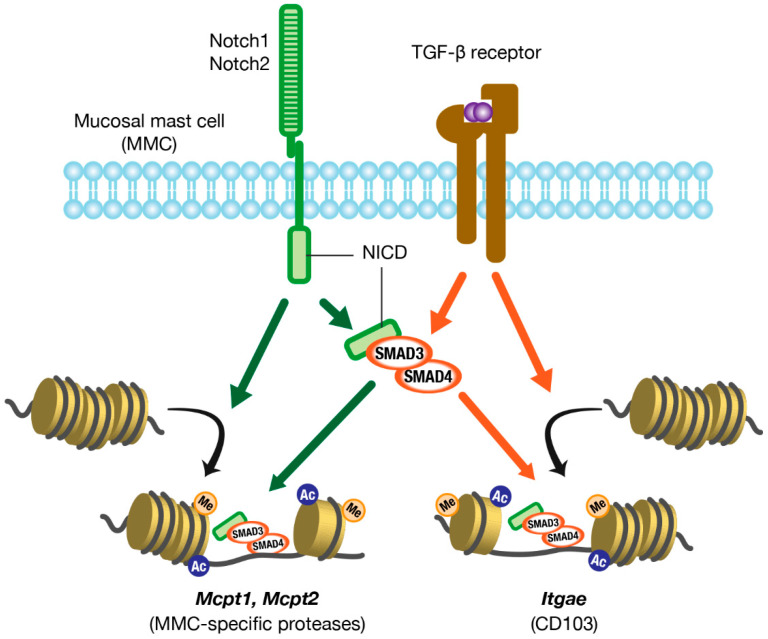
Molecular mechanisms of MMC marker gene expression. Notch and TGF-β signaling act synergistically and interdependently in the expression of MMC marker genes. MMC, mucosal mast cell; TGF-β, transforming growth factor-β; NICD, Notch intracellular domain.

**Table 1 cells-11-00329-t001:** Characteristics of mast cell subtypes.

	Mucosal-Type	Connective Tissue-Type	Ref.
**Mouse**	Mucosal Mast Cell (MMC)	Connective Tissue Mast Cell (CTMC)	
Size	Smaller (5–10 μm)	Larger (10–20 μm)	[17]
Protease	mMCP-1 ^1^, mMCP-2	mMCP-4, mMCP-5, mMCP-6, mMCP-7 CPA3 ^2^	[18]
Proteoglycan	Chondroitind sulfate di-B, A, E	Chondroitind sulfate E Heparin	[19]
Biogenic Amine	Histamine (Low, <1 pg per cell) Serotonin	Histamine (High, >15 pg per cell) Serotonin	[19]
Cell surface Marker Specific Marker	FcεRI, c-Kit/CD117, ST2 αE integrin/CD103	FcεRI, c-Kit/CD117, ST2 Mrgprb2 ^3^	[3,18,20]
Life Span	Few weeks (~40 days)	9–18 months	[21]
Progenitor	Bone marrow-derived β7 integrin^+^ MCp ^4^	Fetal-derived β7 integrin^+^ MCp	[22,23,24]
**Human**	MC_T_	MC_TC_	
Protease	Tryprase	Tryptase, Chymase CPA3, Cathepsin G	[25]
Proteoglycan	Heparin	Heparin	[18,26]
Progenitor	CD34^+^β7 integrin^+^ MCp	CD34^+^β7 integrin^+^ MCp	[27,28]

^1^ mMCP, mouse mast cell protease; ^2^ Carboxypeptidase A3; ^3^ Mrgprb2, Mas-related G protein-coupled receptor B2; ^4^ MCp, mast cell progenitor.

## Data Availability

Not applicable.

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
