# Peer review of "Mucosal Mast Cells as Key Effector Cells in Food Allergies"

_cells, 2022, doi:10.3390/cells11030329_

Round 1

Reviewer 1 Report

The authors have written a scientifically sound review on mast cells and their importance and biological effects in food allergies. They report on murine experimental data and current knowledge in humans.

However, there are some issues that may be solved and may improve the manuscript.

1) As MCs in humans are important effector cells in food allergy the authors should also report on clinical tests to detect the involvment of mast cells. Current clinical tests such as tryptase, MC-specific cytokines (if any), serotonin, histamine in blood, Methylhistamine in urine, leukotrienes/prostaglandine assessment should be discussed in brief. 

2) As human mucosal mast cells have been described scientifically correctly the authors may also report and discuss normal mast cell infiltrations in human adults and children (e.g. data from Ehrsam C et al. very recently published 2021 in JPGN). See also page 3, lines 83-85. 

3) Additional information on MCs involved in non-IgE or mixed (IgE and non-IgE mediated) diseases may be mentioned and discussed (line 29, page1). In line with this I would recommend to briefly mention MC activation triggered e.g. by cytokines or prostaglandins and/or leukotrienes, or e.g. by excipients as a matter of pseudoallergic reactions.  

Author Response

We wish to express our appreciation to the reviewer for insightful comments on our article. These comments have helped us improve our article. Our responses to the Reviewer’s comments are as follow:

  1. As MCs in humans are important effector cells in food allergy the authors should also report on clinical tests to detect the involvment of mast cells. Current clinical tests such as tryptase, MC-specific cytokines (if any), serotonin, histamine in blood, Methylhistamine in urine, leukotrienes/prostaglandine assessment should be discussed in brief.

Response: We thank the Reviewer for this pertinent advice. In accordance with this comment, we have added the following subsection.

[Page 4, line 133]

2.4. Biomarkers of Food Allergies Derived from Activated Mast Cells

As mentioned above, multiple mediators are released from activated mast cells, including MMCs, during the effector phase of IgE-mediated food allergies. Some of these mediators are useful as biomarkers for objectively detecting allergic reactions in patients with food allergies. The biomarkers in blood or urine are important for the accurate detection of allergic reactions caused by oral food challenge, which is the gold standard for diagnosis of food allergies, and for the clinical diagnosis of anaphylaxis. Histamine is a typical mediator released by IgE-mediated mast cell degranulation. However, in order to properly measure plasma histamine levels, it is necessary to obtain a blood sample within 15 and 60 minutes after the onset of anaphylactic symptoms. In addition, special handling of the blood sample is required [46]. In contrast, urinary histamine and its metabolite, N-methylhistamine, can be measured in samples obtained 24 h after the onset of symptoms [46,47]. Tryptase is also a useful mediator for diagnosing of anaphylaxis. In particular, serum total tryptase levels are strongly associated with insect sting-triggered anaphylaxis [48,49]. However, tryptase levels are lower in MMCs (or MCT) than in cutaneous and perivascular CTMCs (or MCTC), and thus, anaphylactic reactions and in-creased tryptase levels elicited by food allergies may be weaker than those elicited by insect sting or intravenous drugs [50]. Recently, urinary PGD2 metabolite, 11,15-dioxo-9α-hydroxy-2,3,4,5-tetranorprostan-1,20-dioic acid, was reported to be a useful diagnostic index of food allergy in both mice and humans [51,52]. Moreover, re-cent metabolomic analyses reveal many promising candidates for food allergy biomarkers [53]. Crestani et al. [54] show that multifood allergy is associated with tryptophan, an important metabolite in the production of serotonin, and 1-arachidonoyl-GPD, a metabolite upstream of PAF. These biomarkers are released at different times after activation of mast cells. Therefore, it is desirable to have many biomarkers that can be used for diagnosis.

  1. As human mucosal mast cells have been described scientifically correctly the authors may also report and discuss normal mast cell infiltrations in human adults and children (e.g. data from Ehrsam C et al. very recently published 2021 in JPGN). See also page 3, lines 83-85.

Response: In accordance with the Reviewer’s comment, we have added the following sentences and cited the paper by Ehrsam et al.

[Page 3, line 87] Intestinal mast cells are found throughout the gastrointestinal tract and normally ac-count for 2–3% of mononuclear cells in the lamina propria. Ehrsam et al. [37] have recently reported the number and distribution of mast cells in the pediatric gastrointestinal tract in healthy individuals as well as in patients with food allergies.

  1. Additional information on MCs involved in non-IgE or mixed (IgE and non-IgE mediated) diseases may be mentioned and discussed (line 29, page1). In line with this I would recommend to briefly mention MC activation triggered e.g. by cytokines or prostaglandins and/or leukotrienes, or e.g. by excipients as a matter of pseudoallergic reactions.

Response: We agree with the Reviewer’s comment. In accordance with this comment, we have added the following sentences.

[Page 1, line 32] In some non-IgE-mediated allergies such as pseudo-allergies, mast cell activation also plays an important role in the onset of allergic symptoms [3-5].

[Page 4, line 126] Mas-related G protein-coupled receptor-X2 (MRGPRX2) (murine ortholog, Mrgprb2) is known as a receptor that causes IgE-independent mast cell activation. The receptor recognizes a broad range of cationic ligands including certain types of drugs, host defense peptides such as LL-37, and neuropeptides [3,5]. Importantly, the receptor is expressed on CTMCs, but not on MMCs (Table 1). Thus, Pseudo-allergic reactions triggered by these molecules, even those ingested orally, may differ from IgE-dependent food allergies in the profile of secreted mediators.

We are thankful for the time and energy you expended. We have revised our manuscript to incorporate your feedback and hope that these revisions persuade you to accept our submission.

Reviewer 2 Report

In this paper the authors study mucosal mast cells (MMCs) located in the intestinal mucosa in the allergic process. Furthermore, they discuss the mechanism by which oral immunotherapy, a promising treatment for patients with food allergies, induces a non-response to food allergens and the role of MMCs in this process.

The abstract lacks a conclusion.

Tables and fig. 1 lack legend.

When the role of mast cells in allergic and inflammatory processes is addressed, this must be present the role of anti-inflammatory IL-37 gene expression and both interleukin IL-33 and IL-25 which induce Th2-type cytokine production by various cell types, suggesting that they contribute to development of allergic disorders. Therefore, to make this paper more interesting for the readers of this important journal, the authors should expand a little the discussion on this subject, in order to give a wider view to the reader. Below I list two interesting articles that should be studied, incorporate the meaning and report them briefly in the discussion, and in the list of references.

Neurohormonal markers in chronic rhinosinusitis.

Compton RA, Lonergan AR, Tsillioni I, Conti P, Ronconi G, Lauritano D, Rebeiz EE, Theoharides TC.J Biol Regul Homeost Agents. 2021 May-Jun;35(3):901-908. 

Immunomodulatory effects of IL-33 and IL-25 in an ovalbumin-induced allergic rhinitis mouse model.

Yang C, Chen N, Tang XL, Qian XH, Cai CP.J Biol Regul Homeost Agents. 2021 Mar-Apr;35(2):571-581. 

I believe these suggestions are important for improving this paper. Without these corrections the paper cannot be published. So I recommend minor revision.

Author Response

Thank you for your helpful comments and suggestions on improving our manuscript. We are thankful for the time and energy you expended. Our responses to the Reviewer’s comments are as follow:

  1. The abstract lacks a conclusion.

Response: Thank you for pointing that out. In accordance with this comment, we have added the following sentences.

[Page 1, line 16] Research focusing on MMCs should provide useful information for understanding the underlying mechanisms of food allergies and will further advance the treatment of food allergies.

  1. Tables and fig. 1 lack legend.

Response: Thank you for pointing that out. Their legends are as follows.

[Footer of Table 1] 1mMCP, mouse mast cell protease; 2Carboxypeptidase A3; 3Mrgprb2, Mas-related G protein–coupled receptor B2; 4MCp, mast cell progenitor.

[Page 5, line 208, Legend of Figure 1] Figure 1.Inducers of MMC differentiation and expansion. MMCs differentiate from bone marrow-derived MCps in mucosal tissues. MMC numbers increase rapidly during allergic inflammation. MMC, mucosal mast cell; MCp, mast cell progenitor; TGF-β, transforming growth factor-β; SCF, stem cell factor.

  1. When the role of mast cells in allergic and inflammatory processes is addressed, this must be present the role of anti-inflammatory IL-37 gene expression and both interleukin IL-33 and IL-25 which induce Th2-type cytokine production by various cell types, suggesting that they contribute to development of allergic disorders. Therefore, to make this paper more interesting for the readers of this important journal, the authors should expand a little the discussion on this subject, in order to give a wider view to the reader. Below I list two interesting articles that should be studied, incorporate the meaning and report them briefly in the discussion, and in the list of references.

Response: Thank you for your helpful suggestions. In accordance with this comment, we have added the following sentences and cited the paper by Yang et al.

[Page 8, line 347] IL-33 and IL-25 secreted by various cell types activate Th2-type cytokine production and exacerbate allergic diseases. Neutralization of these cytokines by specific antibodies has been shown to be effective in suppressing the Th2 immune responses [85]. Therefore, these cytokines, which enhance the production of IL-4 and IL-13, may also be potent therapeutic targets.

In addition, we have added the following sentences and cited the paper by Compton et al.

[Page 9, line 373] In addition to the inhibitory receptor, inhibitory cytokines such as IL-37 may also function to suppress excessive activation of mast cells [88]. Thus, these natural inhibitors of immune responses could also be used as therapeutic agents.

Again, thank you for giving us the opportunity to strengthen our manuscript with your valuable comments and queries. We have revised our manuscript to incorporate your feedback and hope that these revisions persuade you to accept our submission.

Reviewer 3 Report

Comments: Mucosal mast cells as key effector cells in food allergies

In this article authors have  focused on - Effect of Mucosal mast cells and its role as effector cells in food allergies . There are few concerns related to this article.

Table 1- Authors have mentioned about Mouse and human mast cell subtype. Mucosal/ Connective tissue type. They should also focus on Molecular level to explore the difference in Mucosal and connective tissue. Eg . any specific receptors on mucosal mast cell that make them more responsible towards fool allergy

In this article Author have only focus on mast cells. However, food energy seems to be neglected area.

How mast cell can behave in response to various Food allergen need to be mentioned. Eg.  What type of mediators (Primary/ secondary/ tertiary) can be released by Mast cell in response to specific Food allergen.

Find number 6.1. Which type of food allergen can be blocked by notch signalling, Need to be mentioned in this section.

Point 3.2. In addition to those cytokines, we previously showed that Notch ligands are potent 156 inducers of MMC differentiation [50] (Figure 1).

Comment_ Figure 1 does not deal with Notch signalling.

Figure 2. Author should write about Notch. Which notch is mentioned in Figure 2. Do all Notch follow same pathway as shown in Figure 2               

Author Response

Thank you for your helpful comments and suggestions on improving our manuscript. We are thankful for the time and energy you expended. Our responses to the Reviewer’s comments are as follow:

  1. Table 1- Authors have mentioned about Mouse and human mast cell subtype. Mucosal/Connective tissue type. They should also focus on Molecular level to explore the difference in Mucosal and connective tissue. Eg . any specific receptors on mucosal mast cell that make them more responsible towards fool allergy.

Response: We thank the Reviewer for this pertinent advice. In accordance with this comment, we have added a row for "Specific marker" to Table 1.

  1. In this article Author have only focus on mast cells. However, food energy seems to be neglected area.

Response: We think this is an important perspective. Since this special issue is focused on mast cells, it is difficult to write much about them, but based on this point, we have added the following sentence.

[Page 1, line 25] However, these approaches create a serious problem with food energy intake [2].

  1. How mast cell can behave in response to various Food allergen need to be mentioned. Eg. What type of mediators (Primary/ secondary/ tertiary) can be released by Mast cell in response to specific Food allergen.

Response: Thank you for your helpful suggestion. To address this suggestion, we have added the following sentences.

[Page 4, line 126] Mas-related G protein-coupled receptor-X2 (MRGPRX2) (murine ortholog, Mrgprb2) is known as a receptor that causes IgE-independent mast cell activation. The receptor recognizes a broad range of cationic ligands including certain types of drugs, host defense peptides such as LL-37, and neuropeptides [3,5]. Importantly, the receptor is expressed on CTMCs, but not on MMCs (Table 1). Thus, Pseudo-allergic reactions triggered by these molecules, even those ingested orally, may differ from IgE-dependent food allergies in the profile of secreted mediators.

  1. Find number 6.1. Which type of food allergen can be blocked by notch signalling, Need to be mentioned in this section.

Response: In accordance with the Reviewer’s comment, we have added changed the description to show the specific food antigens.

[Page 8, line 310] ... to a mouse model of IgE-meditated food allergy sensitized with ovalbumin (OVA). The expansion of MMCs in the small intestine and colon was significantly suppressed in mice treated with DAPT during the effector phase. Predictably, the severity of allergic diarrhea and degree of hypothermia induced by oral administration of OVA were lower in mice treated with DAPT than in control mice. The concentration of serum OVA-specific IgE in DAPT-treated mice was the same level as in the control mice,

  1. Point 3.2. In addition to those cytokines, we previously showed that Notch ligands are potent 156 inducers of MMC differentiation [50] (Figure 1).

Comment_ Figure 1 does not deal with Notch signalling.

Response: In accordance with the Reviewer’s comment, we have added the description of “(Notch signaling)” to Figure 1.

  1. Figure 2. Author should write about Notch. Which notch is mentioned in Figure 2. Do all Notch follow same pathway as shown in Figure 2.

Response: In accordance with the Reviewer’s comment, we have added the description of “Notch1” and “Notch2” to Figure 2. Our previous studies have shown that Noch1 and Notch2, but not Notch3 and Notch4, are constitutively and highly expressed on mouse mast cells. Both receptors have also been found to be involved in the expression of mucosal mast cell marker genes.

Again, thank you for giving us the opportunity to strengthen our manuscript with your valuable comments and queries. We have revised our manuscript to incorporate your feedback and hope that these revisions persuade you to accept our submission.

Round 2

Reviewer 3 Report

Authors have incorporated the  suggestion so manuscript  be send for publication